# Five-year clinical outcomes of scanning carbon-ion radiotherapy for prostate cancer

**Yosuke Takakusagi**[1,2]*, **Hiroaki Koge**[1], **Kio Kano**[1], **Satoshi Shima**[1], **Keisuke Tsuchida**[1], **Nobutaka Mizoguchi**[1], **Daisaku Yoshida**[1], **Tadashi Kamada**[1], **Hiroyuki Katoh**[1]

1 Department of Radiation Oncology, Kanagawa Cancer Center, Kanagawa, Japan, 2 Department of Radiation Oncology, Yokohama Sakae Kyosai Hospital, Yokohama, Japan

* y-takakusagi@kcch.jp

**Data Availability Statement:** All relevant data are within the paper and its Supporting Information files.

## Abstract

### Background

Carbon-ion radiotherapy (CIRT) has been associated with favorable clinical outcomes in patients with prostate cancer. At our facility, all patients are treated using scanning CIRT (sCIRT). We retrospectively analyzed five-year clinical outcomes of prostate cancer treated with sCIRT to investigate treatment efficacy and toxicity.

### Methods

In this study, we included 253 consecutive prostate cancer patients treated with sCIRT at the Kanagawa Cancer Center from December 2015 to December 2017. The total dose of sCIRT was set at 51.6 Gy (relative biological effect) in 12 fractions over three weeks. We employed the Phoenix definition for biochemical relapse. The overall survival (OS), biochemical relapse-free (bRF) rate, and cumulative incidence of late toxicity were estimated using the Kaplan–Meier method. Toxicity was assessed using the Common Terminology Criteria for Adverse Events version 4.0.

### Results

The median age of the patients was 70 years (range: 47–86 years). The median follow-up duration was 61.1 months (range: 4.1–80.3 months). Eight (3.2%), 88 (34.8%), and 157 (62.1%) patients were in the low-risk, intermediate-risk, and high-risk groups, respectively, according to the D'Amico classification system. The five-year OS and bRF were 97.5% and 93.3%, respectively. The five-year bRF rates for the low-risk, intermediate-risk, and high-risk groups were 87.5%, 93.7%, and 93.4%, respectively (p = 0.7215). The five-year cumulative incidence of Grade 2 or more late genitourinary and gastrointestinal toxicity was 7.4% and 1.2%, respectively.

### Conclusion

The results of this study show that sCIRT has a favorable therapeutic effect and low toxicity in the treatment of prostate cancer.

**Funding:** Research funding from Toshiba Energy Systems and Solutions Corporation (Kanagawa, Japan). The funders had no role in study design, data collection and analysis, decision to publish, or preparation of the manuscript.

**Competing interests:** Hiroyuki Katoh and Daisaku Yoshida received research funding from Toshiba Energy Systems and Solutions Corporation (Kanagawa, Japan). This does not alter our adherence to PLOS ONE policies on sharing data and materials.

**Abbreviations:** RT, radiotherapy; CIRT, carbon-ion radiotherapy; RBE, relative biological effect; sCIRT, scanning carbon-ion radiotherapy; pCIRT, passive carbon-ion radiotherapy; CTV, clinical target volume; PTV, planning target volume; PSA, prostate-specific antigen; CT, computed tomography; OS, overall survival; bRF, biochemical relapse-free; CI, confidence interval; GU, genitourinary; GI, gastrointestinal.

## Introduction

Prostate cancer is the second most prevalent and fifth most deadly cancer in the world [1]. Radiotherapy (RT) is one of the definitive treatment modalities for prostate cancer [2]. Advances in RT technology have resulted in increments in the doses delivered to target volumes without increasing doses delivered to surrounding normal organs [3]. In particular, carbon-ion RT (CIRT) has physical and biological advantages over conventional X-ray RT. Regarding the physical aspect, CIRT offers high dose concentrations to target volumes due to the Bragg peak and a sharp penumbra [4, 5]. Moreover, the relative biological effect (RBE) is approximately three times higher than that of X-rays in the biological aspect [6].

CIRT for prostate cancer was first initiated in 1995 [7]. Since then, several studies have reported favorable clinical outcomes of CIRT for prostate cancer [7–16]. CIRT was initiated at our institution in 2015, and all patients are treated with scanning CIRT (sCIRT) [17]. Compared to conventional passive CIRT (pCIRT), sCIRT offers better dose distribution to the target volume and also lower doses to the surrounding normal organs [18]. Most previous studies have employed pCIRT, and only a few studies contain clinical results of sCIRT [12, 14, 15]. We previously reported preliminary results of sCIRT for prostate cancer; however, the observation period was not sufficient [14]. Recent advances in CIRT technology have led to a shift from pCIRT to sCIRT. However, the safety and efficacy of sCIRT have not yet been fully investigated.

In this study, we further followed the preliminary sCIRT results of the participants of our previous study [14] and retrospectively investigated the long-term clinical outcomes. This study aimed to evaluate the efficacy and safety of sCIRT for prostate cancer.

## Materials and methods

### Patients

The candidates of this study are the same as those in our previous study [14]. Patients with prostate cancer who commenced sCIRT at our institution between December 2015 and December 2017 were included in this study. Our eligibility criteria were as follows: (i) a histopathological diagnosis of prostate adenocarcinoma, (ii) cT1bN0M0 to cT3bN0M0 according to the 7th UICC classification, (iii) performance status of 0–2, (iv) age of >20 years, (v) no prior treatment except for androgen deprivation therapy (ADT). Clinical data were collected in December 2022. Written informed consent was obtained from all patients. The Kanagawa Cancer Center institutional review board approved this study (approval number:2022–114).

### sCIRT

Details of sCIRT techniques are presented in our previous study [14]. The gross tumor volume was not identified. The clinical target volume (CTV) included the entire prostate and proximal seminal vesicles. In T3b cases, the entire ipsilateral seminal vesicle was included in CTV. Prophylactic irradiation of the pelvic region was not performed as in previous studies [7–16]. Planning target volume (PTV) 1 was created by adding 10 mm anterior and lateral to the CTV and 5 mm cephalad and posterior to it. PTV2 was shrunk from PTV1, and the posterior edge of PTV2 was set in front of the anterior wall of the rectum. The total dose was 51.6 Gy (RBE) in 12 fractions; 34.4 Gy (RBE) in 8 fractions for PTV1 and 17.2 Gy (RBE) in 4 fractions for PTV2. The rectum was delineated as the organ at risk from 10 mm above the upper margin of the PTV to 10 mm below the lower margin of the PTV. The dose constraint for the rectum was set at the volume irradiated with 80% of the prescribed dose of <10 cc. sCIRT was performed in all patients.

### Follow-up

After the completion of sCIRT, patients were followed up by a radiation oncologist and an urologist every three months for up to three years after the initiation of CIRT, every six months for 3–5 years, and every year thereafter. Prostate-specific antigen (PSA) levels were measured at each visit. We employed the Phoenix definition for biochemical relapse [19]; i.e., an increase of ≥2 ng/ml from the nadir PSA was considered a biochemical relapse. Imaging studies such as computed tomography (CT), magnetic resonance imaging, bone scintigraphy, and 18F-fluorodeoxyglucose-positron emission tomography/CT were considered when biochemical relapse was detected. Clinical recurrence was diagnosed by a radiation oncologist and an urologist. Toxicity was assessed using the Common Terminology Criteria for Adverse Events version 4.0. Toxicity after three months of treatment was defined as late toxicity, and the worst toxicity grade was recorded as the final grade of toxicity. The period of observation and the time to event were calculated from the date of sCIRT initiation.

### Statistical analysis

The Kaplan–Meier method was used to estimate the overall survival (OS), biochemical relapse-free (bRF) rate, and cumulative incidence of late toxicity. The bRF for each risk group was tested using the log-rank test. A p-value of <0.05 was considered statistically significant. Statistical analyses were performed using STATA (version 17.0, College Station, TX, USA).

## Results

### Patient characteristics

Table 1 summarizes the characteristics of our study participants.

We included 253 patients in this study. The median follow-up duration was 61.1 (range: 4.1–80.3) months. The median age was 70 (range: 47–86) years. According to the D'Amico classification, 8 (3.2%), 88 (34.8%), and 157 (62.1%) patients were classified into the low-risk, intermediate-risk, and high-risk groups, respectively.

### OS

Fig 1 shows the OS, which was 97.5% [95% confidence interval (CI): 94.5%–98.9%] at 5 years.

Eight deaths occurred during the observation period, all from other causes. Six of the deaths were due to other cancers: two from liver cancer, one from lung cancer, one from esophageal cancer, one from stomach cancer, and one from pancreatic cancer, all of which were newly-diagnosed malignancies after the completion of sCIRT. The other two cases were interstitial pneumonia and senility (one each).

### bRF rate

Fig 2 shows the bRF rate for all patients. The five-year bRF rate was 93.3% (95% CI: 89.1%–95.9%).

The classification of the five-year bRF rate by risk group is shown in Fig 3.

The five-year bRF rates were 87.5% (95% CI: 38.7%–98.1%), 93.7% (95% CI: 85.4%–97.3%), and 93.4% (95% CI: 87.7%–96.5%) for the low-risk, intermediate-risk, and high-risk groups, respectively (p = 0.7215). During the observation period, PSA elevation with the nadir plus of >2 ng/ml was observed in 2, 13, and 13 patients in the low-risk, intermediate-risk, and high-risk groups, respectively. Of these patients with the PSA nadir plus of >2ng/ml, 1, 8, and 3 patients in the low-risk, intermediate-risk, and high-risk groups, respectively, experienced spontaneous reductions in PSA levels without any treatment; therefore, these patients were

**Table 1. Patient characteristics.**

|  | n (%) |
|---|---|
| Follow-up duration, months, median (range) | 61.1 (4.1-80.3) |
| Age, years, median (range) | 70 (47-86) |
| T stage |  |
| 1c | 49 (19.4%) |
| 2a | 79 (31.2%) |
| 2b | 35 (13.8%) |
| 2c | 53 (20.9%) |
| 3a | 27 (10.7%) |
| 3b | 10 (4.0%) |
| Pretreatment PSA, ng/ml, median (range) | 8.6 (3.33-187) |
| < 10 | 147 (58.1%) |
| 10 ≤ 20 | 73 (28.9%) |
| 20 ≤ | 33 (13.0%) |
| Gleason score |  |
| 6 | 14 (5.5%) |
| 7 | 117 (46.2%) |
| 8 | 79 (31.2%) |
| 9 | 43 (17.0%) |
| 10 | 0 (0.0%) |
| D'Amico classification |  |
| low | 8 (3.2%) |
| intermediate | 88 (34.8%) |
| high | 157 (62.1%) |
| Androgen deprivation therapy |  |
| none | 9 (3.6%) |
| neoadjuvant | 87 (34.4%) |
| neoadjuvant and adjuvant | 157 (62.1%) |

PSA: prostate specific antigen

excluded from the biochemical recurrence. Thus, the numbers of cases of biochemical recurrence were 1, 5, and 10 in the low-risk, intermediate-risk, and high-risk groups, respectively. Clinical recurrence was observed in three cases. Clinical recurrence was observed 36.6, 47.1, and 56.5 months after the initiation of sCIRT, and recurrence occurred in pelvic lymph node with lung metastases, bone metastases, and local recurrence with para-aortic lymph node metastases, respectively. Three patients with clinical recurrence, who all had Gleason scores of 9, were classified in the high-risk group.

## Toxicity

Table 2 summarizes late toxicity.

Grades 1, 2, and 3 late genitourinary (GU) toxicity were observed in 56 (22.1%), 16 (6.3%), and 1 (0.4%) patients, respectively. The cumulative incidence of Grade 2 or more late GU toxicity is shown in Fig 4.

The five-year cumulative Grade 2 or more late GU toxicity rate was 7.4% (95% CI: 4.7%–11.5%). Patients with Grade 3 late GU toxicity who had been taking anticoagulants prior to treatment showed gross hematuria 50.8 months after the initiation of sCIRT and were hospitalized and treated with hemostatic agents and continuous bladder irrigation; however, no

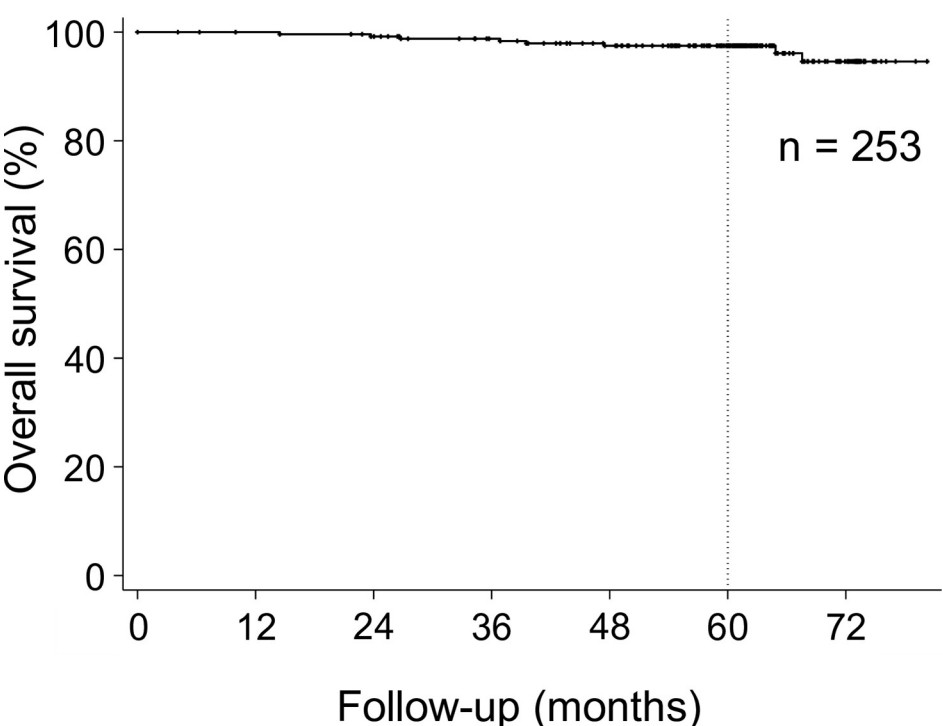

**Fig 1. Overall survival (OS).** The OS at five years was 97.5% (95% CI: 94.5%–98.9%).

transfusion was administered. The cystoscopy performed when treating hematuria revealed radiation cystitis. No Grade 4 or more late GU toxicity was observed.

Grades 1 and 2 late gastrointestinal (GI) toxicity was observed in 12 (4.7%) and 3 (1.2%) patients, respectively. Grade 3 or more late GI toxicity was not observed. The cumulative incidence of Grade 2 or more late GI toxicity is shown in Fig 5.

The five-year cumulative Grade 2 or more late GI toxicity rate was 1.2% (95% CI: 0.4%–3.7%).

Other cases of toxicity included two cases of Grade 1 dermatitis (both of which resolved spontaneously) and a Grade 3 fracture of the right femoral condyle, which occurred 14.4 months after the initiation of sCIRT, following a fall and was managed surgically.

## Discussion

We investigated the five-year clinical outcomes of sCIRT for prostate cancer. The five-year OS and bRF rates were 97.5% and 93.3%, respectively, and only a few serious cases of late toxicity were observed.

Several studies have reported favorable outcomes with CIRT for prostate cancer. The five-year bRF rates/biochemical relapse-free survival rates ranged from 82.6% to 92.7% [7, 8, 11–13, 15]. In a Japanese multi-institutional study on CIRT for prostate cancer, the five-year biochemical relapse-free survival rates were 92%, 89%, and 92% in the low-risk, intermediate-risk, and high-risk groups, respectively [12]. Regarding the five-year bRF rates by risk group, Kawamura et al. reported 91.7%, 93.4%, and 92.0% in the low-risk, intermediate-risk, and high-risk groups, respectively [13], while Sato et al. reported 95.1%, 90.9%, and 91.9% in the low-risk, intermediate-risk, and high-risk groups, respectively [15]. Our study produced comparable results to previous studies. In the present study, the five-year bRF tended to be slightly lower in

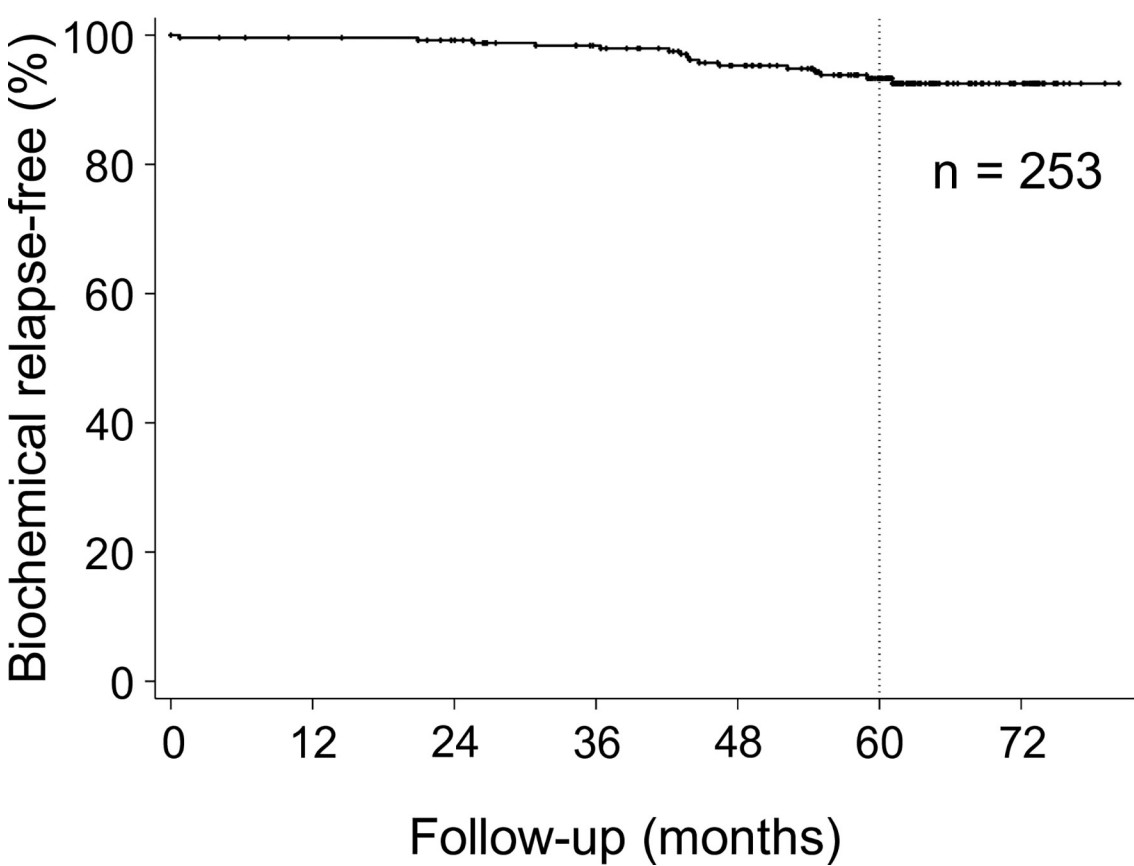

**Fig 2. Biochemical relapse-free (bRF) rates of all patients.** The five-year bRF rate was 93.3% (95% CI: 89.1%–95.9%).

the low-risk group (87.5%); however, since the number of cases in the low-risk group was small, it was difficult to assess the results accurately.

In this study, there were several cases of PSA elevation with the nadir PSA plus 2 ng/ml that met the Phoenix definition in which the PSA levels spontaneously decreased. This transient increment in the serum PSA level, which is known as the PSA bounce, typically occurs after brachytherapy for prostate cancer [20]. PSA bounce has also been reported after stereotactic body radiotherapy for prostate cancer [21]. Currently, there are only a few reports on PSA kinetics after CIRT. Darwis NDM et al. reported that the PSA bounce was observed in 55.7% of patients, out of which 97.6% experienced an increase of less than 2.0 ng/ml [22]. In our previous study, we investigated PSA kinetics after the combination of neoadjuvant ADT and sCIRT in 85 patients with prostate cancer in the intermediate-risk group [23]. In that study, PSA elevations above the nadir PSA plus 2 ng/ml occurred in eight cases, of which seven experienced spontaneous declines in the PSA level. There were several cases of transient PSA elevation above the nadir plus 2 ng/ml in this study as well, and attention should be paid to PSA kinetics after CIRT. The cases judged to have experienced biochemical recurrence in the present study also included several cases in which salvage ADT was initiated immediately upon recognition of the nadir PSA level plus 2 ng/ml. These cases may have also included transient elevations of PSA levels. Further analyses of PSA kinetics after CIRT are needed.

Minimal late toxicity has been reported with CIRT for prostate cancer; late GU toxicity and GI toxicity of Grade 2 or more have been reported to be 6.0%–9.3% and 0.3%–2.4%,

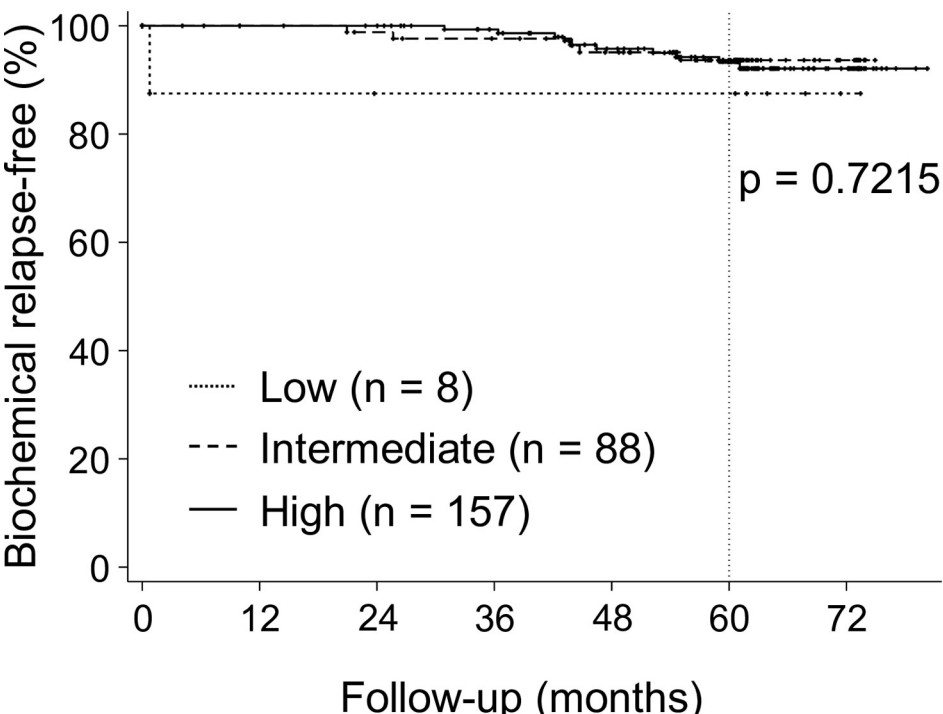

**Fig 3. Biochemical relapse-free (bRF) rates by risk group.** The five-year bRF rates were 87.5% (95% CI: 38.7%–98.1%), 93.7% (95% CI: 85.4%–97.3%), and 93.4% (95% CI: 87.7%–96.5%) for the low-risk, intermediate-risk, and high-risk groups, respectively (p = 0.7215).

respectively [8, 10, 12, 13, 15]. In this study, late GU toxicity and GI toxicity of Grade 2 or more were 6.7% and 1.2%, respectively, which were similar to those reported in previous studies. Compared with our previous study [14], late GU toxicity increased by 4 and 1 cases for Grade 1 and Grade 3, respectively. Of these, Grade 1 GU toxicity was frequent urination in three patients and urinary incontinence in one patient. There was one case of Grade 3 hematuria as mentioned in the results section. Ishikawa et al. revealed that more than 80% of rectal bleeding after CIRT occurs within two years and can occur even after three years of treatment [24]. On the other hand, in this study, more than 90% of rectal bleeding appeared by 2 years, with no new cases occurring after three years. A Japanese multi-institutional study of CIRT for prostate cancer demonstrated long-term toxicity, with GI toxicity showing no new toxicity after five years of treatment and GU toxicity showing a gradually-increasing incidence of toxicity after five years [12]. Therefore, longer follow-up is needed to clarify long-term toxicity.

There are several limitations to this study. First, it was a single-center, retrospective study. Second, the number of patients in the low-risk group was small, making it difficult to accurately assess treatment outcomes in this group. Third, the follow-up and treatment methods after PSA elevation were not defined, and it is possible that patients with transient PSA

**Table 2. Late toxicity.**

|  | Grade 0 | Grade 1 | Grade 2 | Grade 3 | Grade 4, 5 |
|---|---|---|---|---|---|
| Genitourinary | 174 (68.8%) | 56 (22.1%) | 16 (6.3%) | 1 (0.4%) | 0 |
| Gastrointestinal | 238 (94.1%) | 12 (4.7%) | 3 (1.2%) | 0 | 0 |
| Other | 249 (98.4%) | 2 (0.8%) | 0 | 1 (0.4%) | 0 |

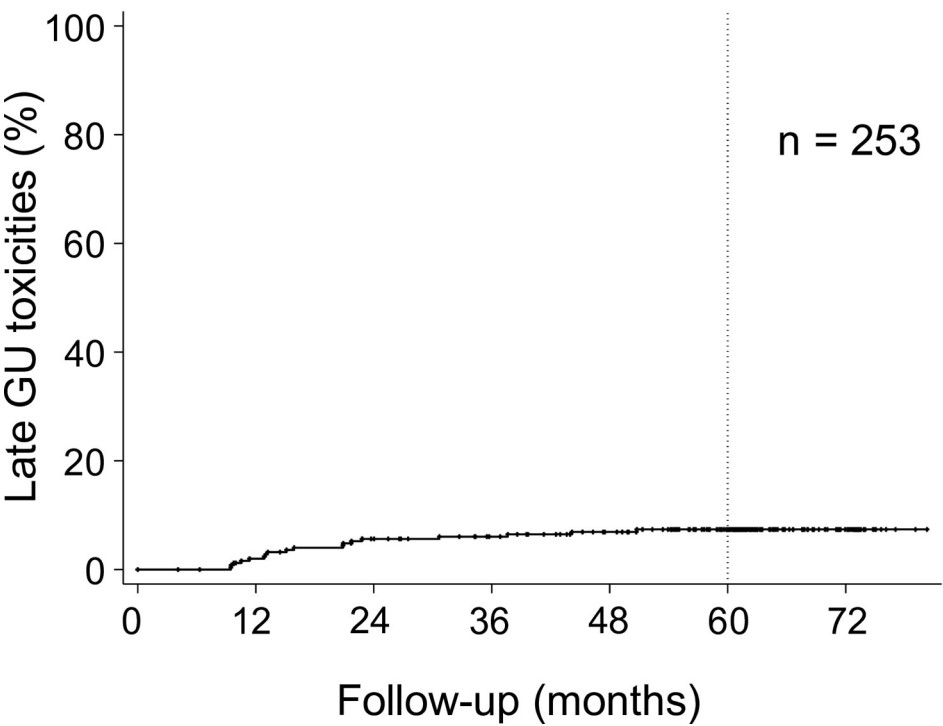

**Fig 4. Cumulative incidence of Grade 2 or more late genitourinary (GU) toxicity.** The five-year cumulative late GU toxicity rate was 7.4% (95% CI: 4.7%–11.5%).

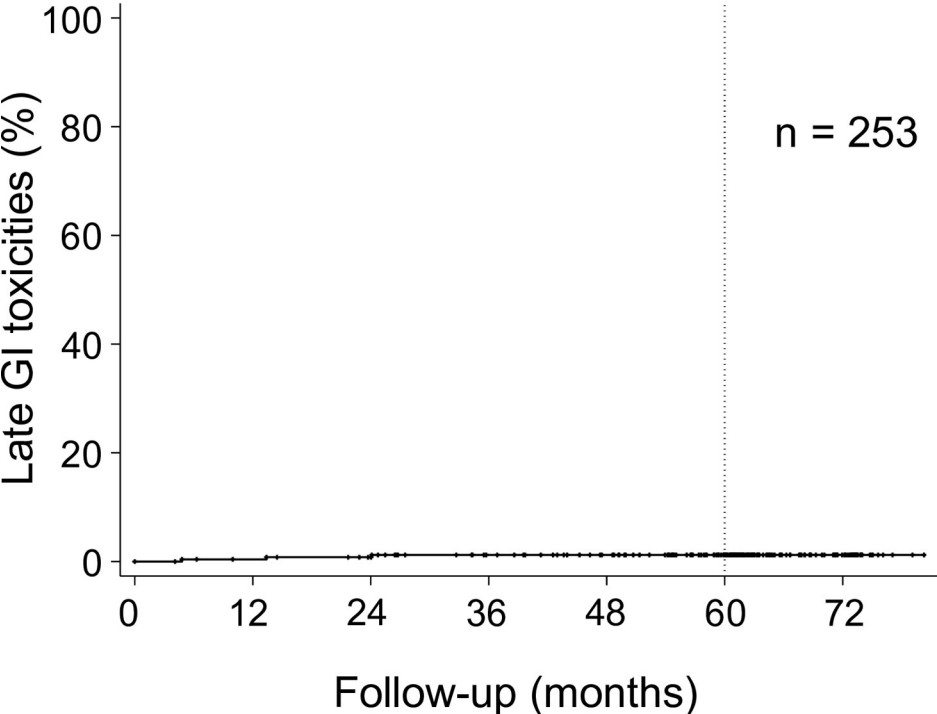

**Fig 5. Cumulative incidence of Grade 2 late gastrointestinal (GI) toxicity.** The five-year cumulative late GI toxicity rate was 1.2% (95% CI: 0.4%–3.7%).

elevation were included among those diagnosed with biochemical recurrence. Further studies on PSA kinetics after CIRT are needed. Fourth, regarding the follow-up period, this study revealed a five-year clinical outcome; however, studies with 10-year clinical outcomes showed increased biochemical recurrence rates and GU toxicity compared to the five-year time point [12]. Therefore, a longer follow-up period is necessary. Fifth, RT is one of the definitive treatments for prostate cancer, however, the results of conventional RT with X-rays and CIRT in this study cannot be directly compared because of differences in dose fractionation and dose constraints.

## Conclusion

sCIRT demonstrated good therapeutic efficacy and low toxicity over a five-year follow-up period. Further follow-up is needed to clarify longer-term clinical outcomes.

## Supporting information

**S1 Data.**
(XLSX)

## Author Contributions

**Conceptualization:** Yosuke Takakusagi, Hiroyuki Katoh.

**Data curation:** Yosuke Takakusagi, Hiroaki Koge, Kio Kano, Satoshi Shima.

**Formal analysis:** Yosuke Takakusagi.

**Investigation:** Yosuke Takakusagi, Hiroaki Koge, Kio Kano, Satoshi Shima.

**Methodology:** Yosuke Takakusagi, Keisuke Tsuchida.

**Project administration:** Yosuke Takakusagi, Daisaku Yoshida.

**Supervision:** Tadashi Kamada, Hiroyuki Katoh.

**Validation:** Nobutaka Mizoguchi.

**Writing – original draft:** Yosuke Takakusagi.

**Writing – review & editing:** Keisuke Tsuchida, Nobutaka Mizoguchi, Daisaku Yoshida, Hiroyuki Katoh.

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
