## [Decision Letter · Decision Letter 0]

21 Jun 2023

PONE-D-23-02959Five-year Clinical Outcomes of Scanning Carbon-ion Radiotherapy for Prostate CancerPLOS ONE

Dear Dr. Takakusagi,

Thank you for submitting your manuscript to PLOS ONE. After careful consideration, we invite you to submit a revised version of the manuscript that addresses the points raised during the review process.

We look forward to receiving your revised manuscript.

Kind regards,

Calogero Casà

Academic Editor

PLOS ONE

Journal Requirements:

2. Please make sure that all information entered in the 'Ethics Statement' section regarding ethics approval and informed participant consent is also included in the Methods section of the manuscript

https://ro-journal.biomedcentral.com/counter/pdf/10.1186/s13014-020-01575-7.pdf

In your revision ensure you cite all your sources (including your own works), and quote or rephrase any duplicated text outside the methods section. Further consideration is dependent on these concerns being addressed.

4. Thank you for including your ethics statement:  "Written informed consent was obtained from all patients. The hospital’s institutional review board approved this study (approval number:2022-114)." 

For studies reporting research involving human participants, PLOS ONE requires authors to confirm that this specific study was reviewed and approved by an institutional review board (ethics committee) before the study began. Please provide the specific name of the ethics committee/IRB that approved your study, or explain why you did not seek approval in this case.

Hiroyuki Katoh and Daisaku Yoshida received research funding from Toshiba Energy Systems and Solutions Corporation (Kanagawa, Japan).

6. Thank you for stating the following in the Competing Interests/Financial Disclosure * (delete as necessary) section: 

Hiroyuki Katoh and Daisaku Yoshida received research funding from Toshiba Energy Systems and Solutions Corporation (Kanagawa, Japan).

We note that you received funding from a commercial source: Toshiba Energy Systems and Solutions Corporation

Additional Editor Comments:

Dear Authors

Thank you for submitting your manuscript to PLOS ONE. After careful evaluation, we believe that the manuscript has all the elements to meet, after minor revision, the publication criteria of PLOS ONE in a minimally modified form. Therefore, we invite you to submit a revised version of the manuscript that addresses the points raised during the review process.

We look forward to receiving your revised manuscript.

Kind regards,

Calogero Casà

Academic Editor

PLOS ONE

Reviewers' comments:

Reviewer's Responses to Questions

**Comments to the Author**

1. Is the manuscript technically sound, and do the data support the conclusions?

Reviewer #1: Yes

Reviewer #2: Yes

2. Has the statistical analysis been performed appropriately and rigorously? 

Reviewer #1: I Don't Know

Reviewer #2: Yes

3. Have the authors made all data underlying the findings in their manuscript fully available?

Reviewer #1: Yes

Reviewer #2: Yes

4. Is the manuscript presented in an intelligible fashion and written in standard English?

Reviewer #1: Yes

Reviewer #2: Yes

5. Review Comments to the Author

Reviewer #1: This study presents an interesting single-center institutional experience with carbon-ion radiotherapy for prostate cancer. As cancer treatment continues to evolve, this study is a noteworthy addition. I see that the authors previously published preliminary results in 2020 and I agree that long-term follow-up is necessary especially in terms of prostate cancer.

I have few comments.

The authors report that 62.1% of patients were in the high-risk category and were treated for prostate only between 2015-2017.

1- Now according to current standards of care and guidelines (NCCN), prophylactic nodal radiation is recommended for high-risk prostate cancer patients. It is suggested that the authors briefly discuss this issue/limitation in the discussion section, particularly in relation to carbon-ion therapy.

2- Additionally, the discussion section should briefly mention the overall current radiotherapy treatment approach for prostate cancer patients and that caution should be exercised when interpreting the data due to the high degree of heterogeneity between this treatment (carbon ion) and photon therapy (such as direct comparison of fractionation, dose constraints) in the current literature.

Reviewer #2: I congratulate the authors on their project. Their data update confirms their previously published data (2020, Radiation Oncology)

Carbon ion radiotherapy (CIRT) has been used to treat prostate cancer since 1995. To date, only 14 institutes have implemented CIRT worldwide. Notably, this study evaluates the use of CIRT using the spot scanning. Despite the limitations of the study highlighted by the authors (single center retrospectively study, only five years follow-up and small size of low-risk patients), the present study shows a good toxicity profile and a good efficacy of carbon ions for prostate cancer. For these reasons this work could be interesting for the scientific community and could be the starting for prospective trial in this setting.

In Discussion section (lines 220-223) please specify that PSA bounce occurs also post-SBRT.

Reference recommened: “Jiang NY, Dang AT, Yuan Y, Chu FI, Shabsovich D, King CR, Collins SP, Aghdam N, Suy S, Mantz CA, Miszczyk L, Napieralska A, Namysl-Kaletka A, Bagshaw H, Prionas N, Buyyounouski MK, Jackson WC, Spratt DE, Nickols NG, Steinberg ML, Kupelian PA, Kishan AU. Multi-Institutional Analysis of Prostate-Specific Antigen Kinetics After Stereotactic Body Radiation Therapy. Int J Radiat Oncol Biol Phys. 2019 Nov 1;105(3):628-636. doi: 10.1016/j.ijrobp.2019.06.2539.”

6. PLOS authors have the option to publish the peer review history of their article (what does this mean?). If published, this will include your full peer review and any attached files.

Reviewer #1: No

Reviewer #2: No

---

## [Author Response · Author response to Decision Letter 0]

13 Jul 2023

Reviewer #1: 

We thank the reviewer for evaluating our manuscript. According to the suggestion, we revised the manuscript as follows.

“1- Now according to current standards of care and guidelines (NCCN), prophylactic nodal radiation is recommended for high-risk prostate cancer patients. It is suggested that the authors briefly discuss this issue/limitation in the discussion section, particularly in relation to carbon-ion therapy.”

Thank you for your precise comments. We had added the sentence in Materials and Methods section to explain that prophylactic pelvic irradiation was not performed as follows:

“Prophylactic irradiation of the pelvic region was not performed as in previous studies [7-16].” (Lines 87 to 88)

“2- Additionally, the discussion section should briefly mention the overall current radiotherapy treatment approach for prostate cancer patients and that caution should be exercised when interpreting the data due to the high degree of heterogeneity between this treatment (carbon ion) and photon therapy (such as direct comparison of fractionation, dose constraints) in the current literature.”

Thank you for your precise comments. We had added the sentence in Discussion section to explain that it cannot be compared our study with previous studies using conventional X-rays as follows:

“Fifth, RT is one of the definitive treatments for prostate cancer, however, the results of conventional RT with X-rays and CIRT in this study cannot be directly compared because of differences in dose fractionation and dose constraints.” (Lines 264 to 267)

Reviewer #2: 

We thank the reviewer for evaluating our manuscript. According to the suggestion, we revised the manuscript as follows.

“In Discussion section (lines 220-223) please specify that PSA bounce occurs also post-SBRT. Reference recommened: “Jiang NY, Dang AT, Yuan Y, Chu FI, Shabsovich D, King CR, Collins SP, Aghdam N, Suy S, Mantz CA, Miszczyk L, Napieralska A, Namysl-Kaletka A, Bagshaw H, Prionas N, Buyyounouski MK, Jackson WC, Spratt DE, Nickols NG, Steinberg ML, Kupelian PA, Kishan AU. Multi-Institutional Analysis of Prostate-Specific Antigen Kinetics After Stereotactic Body Radiation Therapy. Int J Radiat Oncol Biol Phys. 2019 Nov 1;105(3):628-636. doi: 10.1016/j.ijrobp.2019.06.2539.””

Thank you for your precise comments. We had added the sentence in Discussion section to explain that PSA bounce was also observed after SBRT as follows:

“PSA bounce has also been reported after stereotactic body radiotherapy for prostate cancer [21].” (Lines 224 to 225)

And we have added the reference #21 as your recommendation in Lines 362 to 364

---

## [Decision Letter · Decision Letter 1]

14 Aug 2023

Five-year Clinical Outcomes of Scanning Carbon-ion Radiotherapy for Prostate Cancer

PONE-D-23-02959R1

Dear Dr. Takakusagi,

We’re pleased to inform you that your manuscript has been judged scientifically suitable for publication and will be formally accepted for publication once it meets all outstanding technical requirements.

Kind regards,

Calogero Casà

Academic Editor

PLOS ONE

Additional Editor Comments (optional):

All comments have been considered by the authors, the article is scientifically relevant, the methodology is rigorous, and the results are clearly described. In discussion, the paper is well placed in the context of the international literature. The English language is correct and requires no corrections or editing.

Reviewers' comments:

Reviewer's Responses to Questions

**Comments to the Author**

1. If the authors have adequately addressed your comments raised in a previous round of review and you feel that this manuscript is now acceptable for publication, you may indicate that here to bypass the “Comments to the Author” section, enter your conflict of interest statement in the “Confidential to Editor” section, and submit your "Accept" recommendation.

Reviewer #1: (No Response)

Reviewer #2: All comments have been addressed

2. Is the manuscript technically sound, and do the data support the conclusions?

Reviewer #1: Yes

Reviewer #2: Yes

3. Has the statistical analysis been performed appropriately and rigorously? 

Reviewer #1: I Don't Know

Reviewer #2: Yes

4. Have the authors made all data underlying the findings in their manuscript fully available?

Reviewer #1: Yes

Reviewer #2: Yes

5. Is the manuscript presented in an intelligible fashion and written in standard English?

Reviewer #1: Yes

Reviewer #2: Yes

6. Review Comments to the Author

Reviewer #1: Thanks for adding clarifying statements. Still there is a room for improvement and limitations of carbon ion therapy in terms of pelvic nodal radiation can be briefly discussed.

Reviewer #2: The manuscript is interesting and confirms that Carbon-ion radiotherapy is associated with favorable clinical outcomes in patients with prostate cancer even if a longer follow-up is needed to assess these data. It would be interesting to start a prospective trial for stronger evidence. The requested changes were made. Great work!

7. PLOS authors have the option to publish the peer review history of their article (what does this mean?). If published, this will include your full peer review and any attached files.

Reviewer #1: No

Reviewer #2: No

---

## [Editor Report · Acceptance letter]

28 Feb 2024

PONE-D-23-02959R1 

PLOS ONE

Dear Dr. Takakusagi, 

I'm pleased to inform you that your manuscript has been deemed suitable for publication in PLOS ONE. Congratulations! Your manuscript is now being handed over to our production team.

Kind regards, 

on behalf of

Dr. Calogero Casà 

Academic Editor

PLOS ONE